

# A lack of open data standards for large infrastructure projects hampers social-ecological research in the Brazilian Amazon

Jacy L. Hyde[1,*], Christine Swanson[1,2,*], Stephanie A. Bohlman[1,3], Simone Athayde[4], Emilio M. Bruna[5,6] and Denis R. Valle[1]

[1] School of Forest, Fisheries, and Geomatics Sciences, University of Florida, Gainesville, Florida, United States
[2] Division of Undergraduate Education, U.S. National Science Foundation, Alexandria, Virginia, United States
[3] Smithsonian Tropical Research Institute, Balboa, Ancón, Panama
[4] World Resources Institute, Washington, District of Columbia, United States
[5] Center for Latin American Studies, University of Florida, Gainesville, Florida, United States
[6] Department of Wildlife Ecology & Conservation, University of Florida, Gainesville, Florida, United States
* These authors contributed equally to this work.

Corresponding author
Christine Swanson,
acswanso@nsf.gov

## ABSTRACT

New infrastructure projects are planned or under construction in several countries, including in the bioculturally diverse Amazon, Mekong, and Congo regions. While infrastructure development can improve human health and living standards, it may also lead to environmental degradation, such as deforestation, and social change, such as loss of livelihoods. Accessible, high-quality data about infrastructure projects is essential for both monitoring these projects and studying their social and environmental impacts. As a case study, we investigated the availability and quality of data on infrastructure projects in the Brazilian Amazon by reviewing the academic literature and surveying researchers from the conservation and development community. We used the results of these surveys to identify recommended steps for the gathering, organizing, and sharing of infrastructure data by social-ecological researchers and practitioners. Although data on infrastructure in the Brazilian Amazon were generally available, they were often of poor quality and lacked information critical for monitoring and research. Data were often difficult to find on government and non-government websites as well as reformat, resulting in loss of time and resources for researchers and other stakeholders. Discrepancies between researchers' survey responses on data needs and the types of data used in peer-reviewed articles on infrastructure projects indicate the following information was often missing: geographic extent of the project, construction and operation dates, and project type (*e.g.*, paved *vs.* unpaved road). Including these data in a standardized format, along with making them more readily accessible by hosting them in public repositories and ensuring they are current and comprehensive, would facilitate research and improve planning, decision-making, and monitoring of operational and planned infrastructure projects in Brazil and other developing countries.

# INTRODUCTION

Access to comprehensive, high-quality infrastructure project data is critical to studying, monitoring, and mitigating the social-ecological impacts of infrastructure (*Joppa et al., 2016*). This is particularly important given the millions of roads, dams, hydroways, ports, transmission lines and other major infrastructure projects that are currently operational, under construction, or planned worldwide, including planned massive regional, national, or multi-national infrastructure expansions (*e.g.*, the Initiative for the Integration of the Regional Infrastructure of South America (IIRSA) (http://www.iirsa.org/), and China's Belt and Road Initiative (http://english.gov.cn/beltAndRoad/)). Governments, nongovernmental organizations, and project funders collect and make available such data to monitor compliance and assess environmental impacts (*Ciborra, 2005*). Transparency and accountability resulting from making data available throughout the development process may help minimize inefficiency, corruption, and the mismanagement of public construction projects that have resulted in an annual loss of $4 trillion globally (*Transparency and Accountability Initiative, 2014*). Researchers can help improve estimates of trade-offs and impacts for various project alternatives (*Laurance et al., 2015*), and third parties may bring innovative ideas and solutions to the table (*Janssen, Charalabidis & Zuiderwijk, 2012*). They can also hold the government accountable for including social-environmental variables in licensing or construction decisions and developing adequate consultation and compensation processes for affected populations (*Moran et al., 2018*; *Pereira, 2021*; *Transparency and Accountability Initiative, 2014*). While the expansion and improvement of infrastructure can help increase standards of living and improve human health (*Brenneman & Kerf, 2002*; *Calderón & Servén, 2004*; *Estache, 2003*; *Johansson & Goldemberg, 2002*; *Martínez & Ebenhack, 2008*; *Slough, Urpelainen & Yang, 2015*), large infrastructure projects can also lead to environmental degradation (*Laurance, 2018*; *Laurance & Arrea, 2017*; *Pfaff et al., 2018*; *Souza et al., 2019*) and negatively impact Indigenous peoples and other local communities (*Arrifano et al., 2018*; *Fearnside, 1999*; *Gauthier & Moran, 2018*). For example, while hydroelectric dams installed in the Amazon have led to increased power generation for Brazil, they have also led to altered hydrology and floodplain forest inundation (*Swanson et al., 2021*; *Timpe & Kaplan, 2017*), decline in population of fish and other freshwater organisms (*Duponchelle et al., 2021*), displacement of local communities (*Roquetti et al., 2024*), loss of traditional livelihoods (*Arantes et al., 2022*), and more. These and other social-ecological impacts cannot be properly identified and quantified without information about the infrastructure projects themselves.

Government institutions are typically accountable for providing data that is freely accessible and usable (open data) and information about public and public-private infrastructure projects. Government agencies are also usually responsible for planning projects for licensing or other administrative purposes and for monitoring existing projects

and, thus, should have relevant information about these projects. Many countries have access-to-information laws requiring the release of information to the public (*Kaufmann & Bellver, 2005*; *Relly, 2010*) or have signed transparency pledges. Full implementation of these policies is rare, however, due in part to resource and technological constraints, lack of motivation and capacity among agencies, or unclear designation of responsibility (*Attard et al., 2015*; *Ciborra, 2005*; *Di Ciommo, 2015*; *Janssen, Charalabidis & Zuiderwijk, 2012*; *Wang & Lo, 2016*). Consequently, data about public or public-private infrastructure projects are often unavailable (*Attard et al., 2015*).

Even when infrastructure data are available, they may not be of sufficient quality for specific social-ecological research. For example, datasets may lack relevant attributes, be incomplete, or have insufficient metadata to be used effectively. Task-independent data quality standards, which have been proposed by several entities, apply to datasets independent of research question or usage and focus on the completeness, accuracy, and currency of information (*Open Data Charter, 2015*; *Pipino, Lee & Wang, 2002*; *Vetrò et al., 2016*). These standards do not provide guidance on what information should be included within a dataset. However, for nearly all research projects, data quality is defined as the degree of usefulness in a particular task or context and is highly dependent on the user (*Stvilia et al., 2007*), requiring context-specific content. Consequently, even when data do meet task-independent quality standards, the dataset may still be of limited use because it lacks correct or sufficient information to guide a specific decision or to enable a specific task. For example, basic information about project location or date of construction is not always readily available to researchers, requiring them to invest significant time and resources searching for or collecting these data (*Hyde, Bohlman & Valle, 2018*; *Klarenberg et al., 2019*; *Tucker Lima et al., 2016*).

Accessibility to high quality data is especially important in regions undergoing rapid infrastructure development. As a case study, we chose the Brazilian Amazon, which is culturally and ecologically hyperdiverse—it includes 350 ethnic groups in a population of 28 million people and >250 conservation units compromising 47% of the territory (*Soares-Filho et al., 2023*)—but major infrastructure development plans threaten this diversity (*Athayde et al., 2019*). Expansion of roads, dams, navigable waterways, mining and other infrastructure advance national development goals, but come at the cost of deforestation, loss of traditional livelihoods and other social-ecological impacts. Research on infrastructure impacts, based on accurate, complete data, helps predict, plan for, monitor, and mitigate these impacts. Brazil has a strong legal framework to promote transparency, codifying the right to access information in the 1988 Brazilian Constitution and reinforcing this right with various national and international laws, ordinances, and supporting institutions (https://www.freedominfo.org/2012/05/access-to-public-information-in-brazil-what-will-change-with-law-no-12-5272011/). In 2011, Brazil cofounded the Open Government Partnership (OGP) (www.opengovpartnership.org), which seeks to promote transparency, empower citizens, fight corruption, and harness new technologies to strengthen governance. Unfortunately, most Brazilian government portals are not in compliance with international open government data criteria (*Di Ciommo, 2015*; *Wiedenhöft et al., 2023*). It is also unknown whether data standards are followed

throughout all sectors or if the standards are adequate for conducting meaningful research into social-ecological impacts of development.

Guidelines for the content of infrastructure datasets may improve the usefulness of these datasets for social-ecological research by ensuring they contain certain critical attributes and information (*Joppa et al., 2016*). Using infrastructure development in the Brazilian Amazon as a case-study, we conducted a systematic review of how infrastructure data have been used in social-ecological research in academic publications. Specifically, we asked: (1) What data are required for social-ecological research related to infrastructure projects? (2) How accessible and complete are public datasets on infrastructure projects in the Brazilian Amazon? We then surveyed practitioners and researchers about their data needs and efforts in searching for and using data on infrastructure. We used the results of our literature review and survey to identify what attributes should be included in all infrastructure datasets to maximize the utility of these datasets for researchers and other interested parties. Finally, we evaluated two datasets available on Brazilian government websites to determine how well they conformed to task-independent standards and whether they included the critical attributes we identified. While our study focuses on open data from Brazil, our recommendations are broadly applicable to infrastructure and development projects across the world.

## MATERIALS AND METHODS

Portions of this text were previous published as part of a preprint (*Hyde et al., 2024*).

### Systematic literature review

As a case study for data accessibility and quality worldwide, we chose the Brazilian Amazon. We performed a systematic literature review to determine what information researchers have used when assessing the social and ecological impacts of infrastructure projects in the Brazilian Amazon. We only considered studies published after 2011, the year the Brazilian Federal Access to Information Law (Lei. 12.527/2015) was passed, recognizing that research for studies published early in this period (2011–2013) may have been conducted before the 2011 law was passed. We repeated the analysis excluding the 19% of 62 studies published from 2011 to 2013 to determine whether including these studies might bias our results, and the results were nearly identical as with the full data set. Because it is impossible to know exactly which studies were conducted before the passage of the law, we show the results from all studies from 2011 to 2018. We performed the literature review using the Web of Science's[1] "Core Collection" in September and October 2018. The search strings we used for the literature review are available at https://doi.org/10.5281/zenodo.10626908. We specifically looked for studies focusing on environmental impacts, management, or conservation in relation to current or planned infrastructure projects in the Brazilian Amazon. We only included studies if they specifically used some type of infrastructure dataset or information in their analysis or required infrastructure data to plan the research study. For each study, we determined the type of information used about infrastructure projects and focused on the project attributes (*e.g.*, construction date, location, budget, *etc.*). We recorded the citation, topic, academic discipline,

[1] The Web of Science (WOS), previously known as Web of Knowledge, is an online subscription-based scientific citation indexing service that provides a comprehensive citation search. The Web of Science Core Collection consists of six online databases: Science Citation Index; Social Sciences Citation Index. Arts and Humanities Citation Index; Emerging Sources Citation Index; Book Citation Index; and Conference Proceedings Citation Index. Additional databases available in WOS searches include SciELO Citation Index; BIOSIS Citation Index; MEDLINE1; CABI; and Zoological records. Website: https://clarivate.com/products/web-of-science/ Source: Wikipedia: https://en.wikipedia.org/wiki/Web_of_Science.

infrastructure type, the dataset(s) and types of data used, and the infrastructure attributes for each study.

## Key informant survey

To understand data needs and experiences, we surveyed key researchers and practitioners who focus on social and/or ecological topics from a list of the corresponding authors of the articles in the literature review, members of the Amazon Dams International Research Network (http://amazondamsnetwork.org) (ADN; *Athayde et al., 2019*), and members of the Governance and Infrastructure in the Amazon (https://giamazon.org) (GIA) working group (*Mere-Roncal et al., 2021*). The ADN and GIA coordinate social-ecological research and information-sharing about infrastructure in the Amazon and are comprised of researchers, NGO practitioners, and members of government agencies. With the survey, we collected demographic information and asked participants questions about types of infrastructure projects for which they searched, the information required about these projects, where they searched for information, how long it took to find relevant information and format it for use, what they did if they could not find appropriate data, and about data quality based on task-independent standards (*Vetrò et al., 2016*). Finally, we asked participants to list and rank infrastructure project attributes that were important for their use. We specified that infrastructure projects they were researching could be in any phase of operation (*e.g.*, fully operational, under construction, or in planning). We note that we did not ask for information about the specific states or municipalities within the Brazilian Amazon where research was conducted as we were focusing on data availability for the Brazilian Amazon as a whole. However, other studies have found that data accessibility and transparency can vary by state or municipality (*Kawashita et al., 2024*). This survey was approved by the University of Florida's Institutional Review Boards (IRB #B201600928). Respondents gave written consent to participate in the survey. The full survey is available at https://doi.org/10.5281/zenodo.10626908.

From the survey responses, we summarized which attributes were most important across infrastructure projects. We compared the data survey respondents wanted to data used in the literature and considered discrepancies between the two sources a possible data gap where necessary data might not be available. We evaluated the data quality and the amount of effort spent on data gathering, cleaning, and formatting by performing summary statistics on survey responses. When examining differences in data quality between data retrieved from government *vs.* non-government sources, we only considered answers from participants who reported retrieving data exclusively from a government repository or exclusively from a non-government repository. We also combined responses for all non-government sources (*i.e.*, academic, NGO, other).

## Proposing critical attributes for infrastructure data sets

Based on attributes used in the literature review and survey participants' rankings of attribute importance, we created context-specific standards for infrastructure datasets that are complimentary to the task-independent data quality standards. We considered attributes that were ranked in the top five in the key informant survey more than 40% of

the time as critical for inclusion in infrastructure datasets. By identifying these critical attributes, we strived to encourage the availability of information required to conduct social-ecological research about infrastructure projects.

### Evaluating available infrastructure datasets

To further understand the quality of open data on infrastructure from the Amazon region, we evaluated two infrastructure datasets on whether they contained the attributes we identified as critical for social-ecological research and whether they complied with the task-independent framework provided by *Vetrò et al. (2016)*. Our test cases were large dams and roads, as they are drivers of social-ecological change in the Amazon (*Chen et al., 2015*; *Latrubesse et al., 2017*; *Laurance & Arrea, 2017*; *Nepstad et al., 2001*) and frequently appeared in our survey responses and literature review. Therefore, it is especially important that these data are of high quality and useable for social-ecological research.

We downloaded data on May 22, 2019 from the agencies that oversee the dams and roads: the Agencia Nacional de Energia Elétrica (ANEEL) (https://dadosabertos.aneel.gov.br/) and the Departamento Nacional de Infraestrutura de Transportes (DNIT) (https://www.dnit.gov.br/planejamento-e-pesquisa/dnit-geo), respectively. We chose government datasets because that is where researchers most often look for data. We evaluated the quality of these two publicly available infrastructure datasets based on inclusion of information we identified as critical (see previous paragraph) and five characteristics from *Vetrò et al.*'s *(2016)* task-independent framework: (1) accuracy of spatial components, (2) completeness, (3) currency (up-to-date), (4) machine-readability, and (5) metadata quality. We assessed the spatial accuracy of the datasets by randomly selecting 50 existing projects in each dataset and verifying their locations in Google Earth using the same map projection. If the project was within 30 m (the size of a Landsat pixel) of the location listed on the dataset, it was considered spatially accurate. We quantified how current the dataset was based on the date of the last update. Completeness was difficult to assess because it was unclear in many of the columns whether an empty cell was purposefully empty (the metadata did not provide this information). Instead of scoring the whole data set based on completeness, we chose the first two columns in each data set that were understandable without metadata and that clearly should have been complete, and we determined the percent of empty cells in these columns. Metadata quality was used as proxy for traceability (which measures the history of the data set) and understandability, both of which are somewhat subjective. Thus, we considered the metadata complete if it was present and contained explanations of the attributes in the data, its author, the geographic coordinate system of the shapefile, and the publication date.

## RESULTS

### Systematic literature review

Sixty-two studies fit our criteria for the systematic literature review of articles that have investigated social-ecological impacts of infrastructure in the Brazilian Amazon. The articles focused on a range of topics, most frequently on social issues (such as displacement, livelihoods, socio-environmental conflict, human health, *etc.*), land use/land

cover change (LULCC), and aquatic ecology (Fig. A1A). Together, the articles used infrastructure data 94 times, requiring 236 attributes about those infrastructure projects (Fig. A1B). Hydropower projects were the most common infrastructure category investigated (43 of the 94 datasets), followed by road and highway projects (18 of the 94 datasets). By far the most used attribute about infrastructure projects was the geographic location of the project (66 uses). The project name, the geographic coordinates for the full extent of the project (full geographic extent), and basic technical information were also used frequently (Fig. A1C).

### Key informants survey

From the 472 people we contacted, a total of 87 people responded to the survey, with 68 completions (response rate = 18.4%, completion rate = 14.4%). Most participants (61.8%) were located in Brazil and 70.6% of respondents were in some stage of an academic career (Figs. A2A, A2B). Participants were primarily researching socioeconomic topics (20.6%), land use/land cover change (20.0%), Indigenous peoples (16.3%), and natural resources (13.2%) (Fig. A2C).

There were 539 instances for which data on infrastructure were used by the survey participants. Combined, hydroelectric dams (28.8%), small dams (14.6%), and roads and highways (9.6%) accounted for more than half of the data searches (Fig. A2D). Participants searched for information relatively evenly across project phases: 37.5% searched for information about the planning phase, 34.1% the construction phase, and 28.4% the post-construction phase. Although participants searched for a wide range of information about the infrastructure projects; point location (10.5%), name (8.7%), status (8.4%), construction and operation dates (8.3% each), and full geographic extent (8.3%) were the most sought-after data attributes (Fig. A3). We note that less cited attributes, such as project owner and funding source, may be critical for certain important analyses, such as governance and economic context of infrastructure projects. However, with this manuscript, we intended to focus on availability of the most sought-after infrastructure attributes as a starting point to analyze data accessibility.

### Comparison of attributes in literature review *vs.* survey

There were substantial gaps between the data attributes ranked within the top five needed for research by survey participants compared to the frequency of use these attributes in articles we reviewed (Fig. 1). Most attributes were ranked within the top five attributes required for social-ecological analysis in the survey at a higher rate than they were used in the studies we reviewed, including the geographic extent of the project, construction and operation dates, and project name and status. In contrast, point location was used more often in the literature than it was ranked in the top five attributes, possibly indicating this attribute was more available than the geographic extent of the infrastructure project, which may have been a more useful attribute. Combined, these results highlight potential gaps in infrastructure data availability.

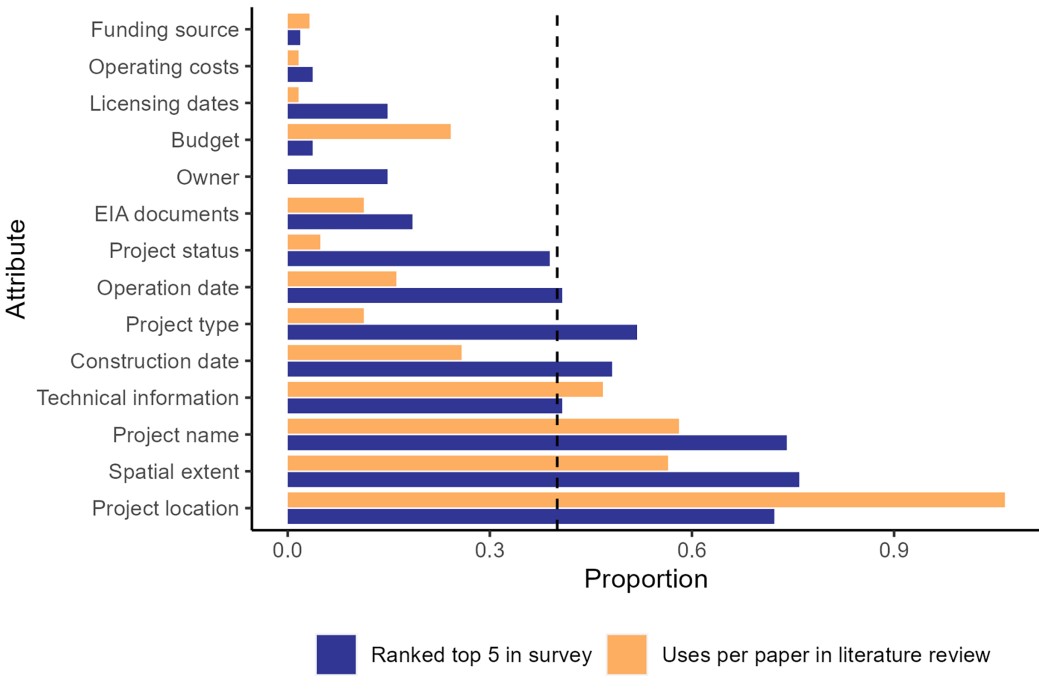

**Figure 1 Column chart of uses of infrastructure data attributes for each article in the literature review.** Orange is the number of times each attribute was used for a unique analysis divided by the number of articles in the literature review conducted in the Web of Science database (WOS) for the 2011–2018 period. Blue is the number of times each attribute was ranked in the top five for importance to include in an infrastructure data set divided by the number of survey participants. EIA stands for environmental impact assessment. The dashed line at 0.4 is the cutoff for the attributes identified as critical for infrastructure datasets.

## Data accessibility and quality

Government sources were the most common place to search for information (39%), but academic and NGO sources were also frequently queried (31% and 24%, respectively) (Fig. 2). Of the 188 searches for government data, 82% of datasets were found from government sources, 14% required additional searches on non-government sources to access the data, and only 3.7% were not found at all. Respondents reported successful access to data from academic and NGO sources in 66% and 64% of the attempts, respectively, a lower rate than from government sources. For academic and NGO sources, 2% were not found (Fig. 2).

Survey participants reported high uncertainty about accuracy of non-spatial components of data. Thirty-nine percent of respondents reported either that non-spatial data had low accuracy or that the respondent could not evaluate the accuracy (Fig. 3A). Conversely, all participants reported that spatial accuracy was at least moderate in quality (Fig. 3A). Accuracy ratings were similar for data obtained from both government and non-government sources (Fig. A4). Respondents rated data sets low for task-independent standards (Fig. 3B). The highest scoring category of task-independent standards was machine readability, although one-third of the data sets scored low in this category. Other categories scored even worse, with 45%, 65% and 88% of the respondents rating

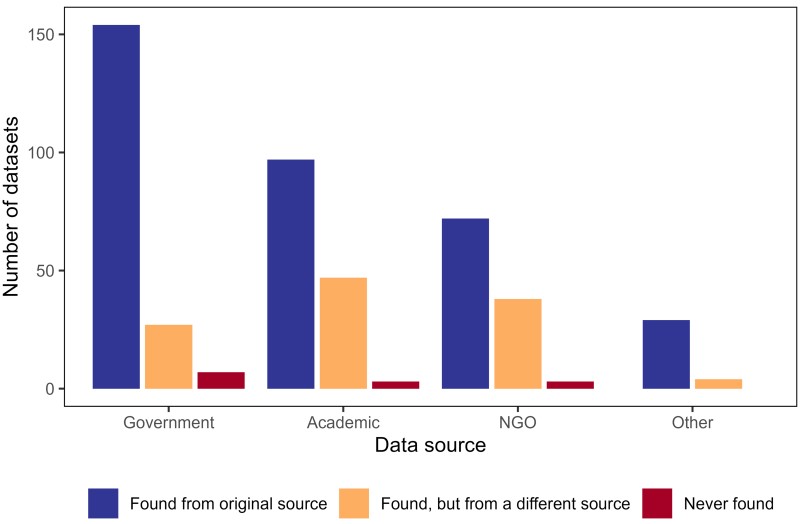

**Figure 2 Column chart displaying number of times survey participants search for and found infrastructure data sets from various sources.** Number of infrastructure data sets that survey participants searched for and found from the data sources in which they originally searched (blue), searched for and found from a different search data source (orange), or searched for but did not find in any data repository (red). Participants may have searched for the same data set from different sources. For example, a participant may have searched for a data source from the government, not found it (yellow bar for government data source), then searched in academic data sources and successfully found the data set there (blue bar for academic data source), but we were unable to track this information.

currentness, metadata quality and completeness, respectively, as low (Fig. 3B). Data acquired from government data sources scored higher in terms of task-independent data quality compared to non-government sources (Fig. A4). For example, almost 75% of the government-sourced data was machine readable, compared to only 40% of non-government-sourced data (Fig. A4).

Respondents reported being able to find data for more than 95% of their searches (Fig. 4). Ten datasets were not found (Fig. 4) for a variety of infrastructure types: large hydroelectric dams (1), railroads (1), roads (1), solar energy plants (1), transmission/distribution lines (1), waterways (1), wastewater/sewage (2), and small dams (2). For the ten unfound datasets, five respondents abandoned their projects altogether, two used proxy datasets, two collected the data themselves, and a third respondent unsuccessfully attempted to collect data.

The time spent searching for data showed a bimodal distribution. For datasets that were found, 35% of respondents reported spending less than 8 h searching for data before finding it, while almost 30% spent more than 168 h in their search for data (Fig. 4). This is the equivalent of more than 1 month's worth of work, assuming a 40-h work week.

Even when respondents were able to find data, 90% of the datasets required additional work to be put into a usable format (Fig. 4). Time spent formatting data also showed a bimodal distribution. Thirty-five percent of datasets required less than 8 h for formatting whereas, 37% needed at least 168 h to be useable (Fig. 4). Twenty datasets (10.6%) never

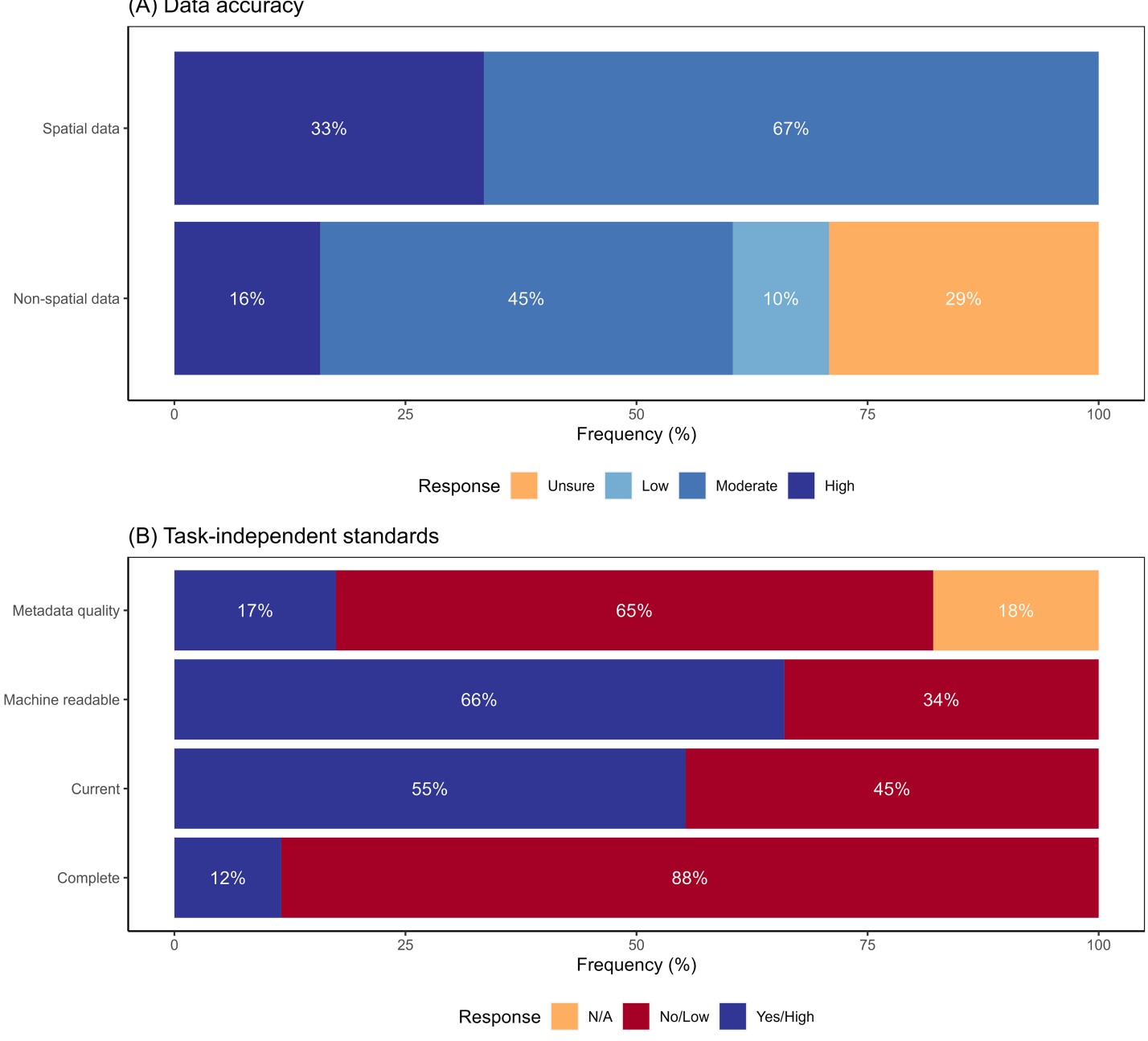

**Figure 3 Survey responses to questions about data quality.** Responses to questions about data quality for: (A) data accuracy (both spatial and non-spatial components); (B) task-independent standards—completeness, currency (<1 year since update), machine readability, and metadata quality. N/A value for metadata quality indicates respondent did not require metadata for the dataset they used.

became usable and varied in quality. Three of these datasets were incomplete, eight were not current, four were not machine readable, and six had low quality metadata. Time researchers spent searching and formatting data was similar between government and non-government sources (Fig. A5).

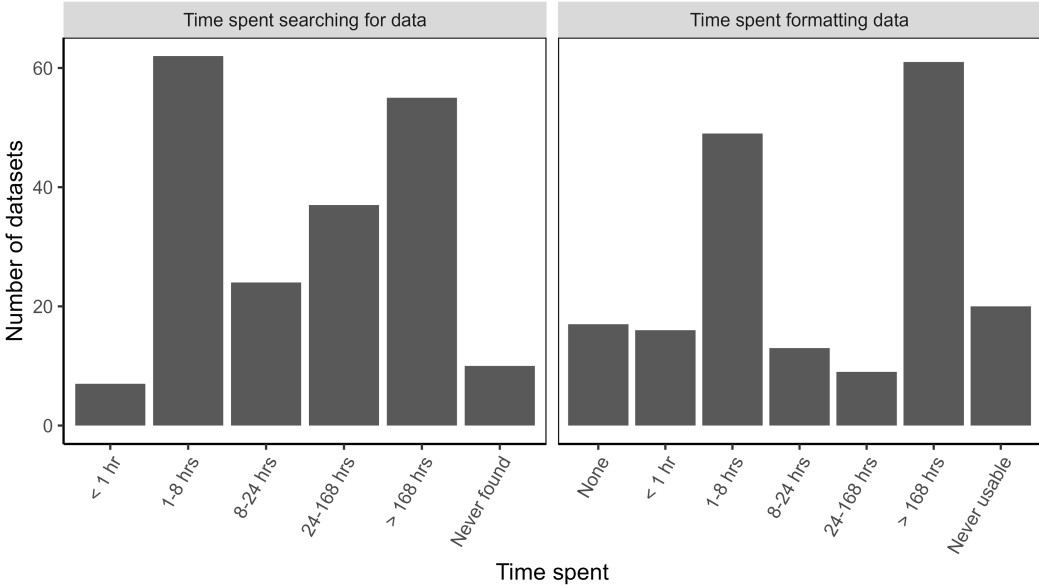

**Figure 4 Time spent searching for and formatting data about large infrastructure projects in the Brazilian Amazon.**

## Critical attributes for infrastructure datasets

Based on the feedback from the literature review and surveys, we propose a list of critical attributes that should be included in all infrastructure datasets. We note that in many countries, including Brazil, this information is required to be provided on a case-by-case basis in the text of environmental impact assessments for individual infrastructure projects. However, these attributes may not be easy to obtain in complete, machine readable formats with confidence in the data accuracy. At a minimum, all infrastructure datasets should include:

(1) The project name

(2) Spatial extent of the project

(3) Basic technical information about the project, which would vary by project but may include capacity (electrical or physical), voltage, bandwidth, number of beds, number of students, *etc.*

(4) Date construction started on the project

(5) Project type, which also varies by infrastructure type but may also include paved *vs.* dirt road, run-of-river *vs.* impoundment dam, primary *vs.* secondary school, highway *vs.* access road, *etc.*

(6) Date project began operations

## Evaluation of public datasets

We evaluated whether two publicly available infrastructure datasets (large hydroelectric dams and roads) contained the critical attributes described in the above section and

assessed the quality of these datasets based on the task-independent standards. Though we were not able to find either dataset on Brazil's central data repository (dados.gov.br), the datasets were available from the websites of the agencies that oversee dams (Agencia Nacional de Energia Elétrica) and roads (Departamento Nacional de Infraestrutura de Transportes). Neither dataset contained all the critical attributes we proposed (Table 1). While both datasets did include project names, neither included project construction or operation dates. The dams dataset included point data and technical information on reservoir sizes but did not include the geographic extent of reservoirs or the dam buildings. The roads dataset included project type (*e.g.*, paved/unpaved, federal/state) and geographic extent in the form of line features for the full length of roads but lacked technical information.

The datasets had high spatial accuracy with 96% (dams) and 98% (roads) of randomly selected points falling within 30 m of their location based on satellite imagery. To measure completeness, we determined the number of filled cells within the first two easily understandable columns: project name and owner for dams, and project name and status for roads. For the dams dataset, 94.5% of cells within these two columns were filled while 100% of the cells within the two roads columns were filled. Across all columns, 17.5% (dams) and 21.7% (roads) of cells were missing data, but it was unclear whether these cells were supposed to be empty because there were no attribute descriptions in the metadata. The dams dataset was current, but there was no information on the currency of the roads dataset (Table 1). Both datasets were available in machine readable formats (ESRI shapefile or KMZ). The metadata quality was low for both datasets. The roads dataset included no metadata for author, date of creation or latest update, attribute description, or geographic datum. The metadata for the dams dataset only included information about the date of latest update (Table 1).

## DISCUSSION

### Data quality and availability for the Brazilian Amazon

We found that infrastructure data for projects in the Legal Amazon were generally available, especially from government sources. However, while about 40% of those surveyed found data with little time investment (<8 h), roughly an equal number of respondents reported that finding these data required extensive, time-consuming searches (>168 h). While some attributes of the data sets, such as spatial accuracy of point locations, were perceived as generally accurate, other attributes of the data sets were perceived as low quality based on task-independent standards, which required many users to spend additional time on data formatting to allow the data to be used for analysis. The low quality of some attributes was exacerbated by low quality of metadata for attribute description and other descriptors of the data. Studies of publicly available data of non-infrastructure data have shown similar issues in accessibility and usability (*Roche et al., 2015*; *Vines et al., 2014*) indicating a broad need for greater data accessibility across disciplines. The federal government is often the main regulator, if not the main funder and co-owner, of large-scale infrastructural projects, and government repositories were the most common place where stakeholders searched for these data. Therefore, it is especially important that government

**Table 1 The quality and content of publicly available infrastructure datasets based on our proposed critical attributes and task-independent standards.**

|  | Federal roads | Large dams |
| --- | --- | --- |
| Dataset origin | Departamento Nacional de Infraestrutura de Transportes | Agencia Nacional de Energia Elétrica |
| Critical attributes |  |  |
| Project Name | Yes | Yes |
| Geographic extent | Yes | No (point location only) |
| Basic technical info | No | Yes (capacity, drainage area) |
| Construction date | No | No |
| Project type | Yes (pavement status, federal status, road size) | No |
| Operation date | No | No |
| Task-independent standards |  |  |
| Spatial accuracy | 98% | 96% |
| Completeness (2 columns) | 100% | 94.3% |
| Currentness | No information | Yes (updated 5/8/19) |
| Machine readable | Yes (shapefile, KMZ) | Yes (shapefile, KMZ) |
| Metadata |  |  |
| Author | No | No |
| Date of creation/update | No | Yes |
| Attribute descriptions | No | No |
| Datum | No | No |

agencies provide easily accessible and high-quality data for projects under their jurisdiction. Unfortunately, our survey reveals that not all government data is easily accessible or interpretable. For example, much of the time participants spent searching for data may have been spent looking across multiple government websites and jurisdictions for desired data and/or parsing difficult-to-navigate websites (*de Oliveira & Silveira, 2018*). A central repository to host infrastructure data, either provided by the federal government or by non-government institutions, would likely reduce the search time thereby increasing the accessibility of information. Examples of non-governmental central repositories that store infrastructure and environmental data include Global Forest Watch and MapBiomas. These and other central repositories could serve to collate data across jurisdictions.

Our results demonstrate some of the costs of poor accessibility and quality. After previously investing time in searching for data (and in some cases, putting them into usable formats), 12 projects were abandoned, presumably due to poor quality or missing data. Alternatively, some researchers invested time and resources to collect new data, thereby having less time and resources for other projects. The cost of poor quality or inaccessible data in time and financial resources are more easily inferred from our survey, but it is challenging to determine the costs associated with failing to generate potentially crucial information for assessing and managing impacts and planning future projects more sustainably. Delays or failures to create this information could have long-term consequences during the global infrastructure boom.

## Gaps in data availability

The discrepancy between the desired attributes (from the survey) and previously used attributes (from the literature review) may indicate an important data accessibility gap, with project type, construction and operation dates, and project status being the least accessible, as represented in published studies, but highly demanded attributes in the surveys. The lack of and/or poor quality of these attributes was confirmed by our analysis of the publicly available roads and dams datasets. Similarly, point location was used far more often than the geographic boundaries of the project in published studies, although both attributes were requested equally in the survey, likely indicating that the full geographic extent of projects is less available. The failure to include critical desired infrastructure information likely impacts the details of research and monitoring being conducted. For example, utilizing dates of construction, operation and changes in project status may allow more nuanced analysis of the timeline of impacts in contrast to using a single date to determine pre- and post-project impacts. Similarly, using geographic extent rather than a point location for an infrastructure project may be important to accurately determine the spatial extent of direct and indirect impacts of that project, such as deforestation, land use change, and displacement. Finally, attributes not at the top of the list for desired attributes, such as project owner and funding source, may be important for specific research and the use of the research for conservation and management.

## Critical data attributes for social-ecological research

Previous studies illustrate how the attributes we identified as being critical for understanding impacts of infrastructure have been used. For example, *Swanson & Bohlman (2021)* used the names, operations dates, dam types (run-of-river *vs.* impoundment), and geographic extent of reservoirs to quantify changes in land cover in the Tocantins River watershed after the installation of multiple hydropower dams. *Nickerson et al. (2022)* compared deforestation surrounding large hydropower dams and small dam clusters in the Legal Amazon, which required information about the construction and operation dates as well as technical information and type of dams. From an energy planning perspective, *de Faria & Jaramillo (2017)* investigated alternatives to hydropower expansion in the Amazon using location and capacity data on all current and planned wind, solar, thermal, and hydropower in the region. Additionally, *de Souza et al. (2023)* used data for highways, waterways, railways, and ports, including project type, geographic extent, and technical information, to model port hinterlands in the Brazilian Amazon. Finally, *Menezes et al. (2018)* modeled vulnerability of Amazonian municipalities to climate change using technical information about hospitals. These studies demonstrate the importance of information about specific infrastructure projects to monitor social-ecological change related to new development and to improve planning and monitoring efforts for future infrastructure development. They also illustrate that the content-specific attributes recommended for inclusion in infrastructure datasets are ubiquitous across infrastructure types.

Though this study focused on the Brazilian Amazon, the information needed from infrastructure data is likely to be the same in other geographic regions. For instance,

*Lupinetti-Cunha et al. (2022)* and *Tisler, Teixeira & Nóbrega (2022)* both used the geographic extent and project type to model the effects of roadless areas on land cover change and conservation across all of Brazil, not just the Amazon. Beyond Brazil, *Flecker et al. (2022)* used location and technical information (capacity in MW) about hydropower dams to investigate how environmental impacts from damming could be reduced across the Amazon basin. *Baird et al. (2021)* used information including dam names, operation and construction dates, spatial extent, and technical information to investigate downstream impacts of dams and the need for Indigenous and traditional knowledge to mitigate those impacts in the Amazon, Canada, Laos, and Vietnam. Finally, *Ding et al. (2022)* used information about project type to model the carbon emissions of 5G cell stations in China. These studies illustrate that the information identified in our literature review and through our key informant survey is applicable beyond the geographic boundaries of the Brazilian Legal Amazon. Though we developed these recommendations based on research conducted in the Brazilian Amazon, our recommendations were designed to be general enough to be applicable to infrastructure development initiatives across the globe and to improve the usability of infrastructure data for a broad range of research initiatives.

## Accuracy of assessed data

Our review of the public data sets for dams and roads from government websites revealed that while Brazil appears to follow the laws regarding access to information by freely providing basic information about infrastructure projects, the overall task-independent and conservation-specific quality could be substantially improved to increase the usefulness of these data. Neither dataset contained all the critical data we identified in our study. Both had low-quality metadata, which made it challenging to interpret many of the attributes. Furthermore, external verification of the data sets was difficult. For example, it was impossible to verify the accuracy of the non-spatial data about the infrastructure projects without exhaustively searching planning documents, a task that would have taken dozens to hundreds of hours to complete. This uncertainty is also reflected in the frequent "I don't know" responses from our survey participants about the accuracy of non-spatial components. Validation of spatial accuracy may be more easily accomplished by using satellite imagery or other remote sensing data, but this validation is still a time-consuming endeavor. As an example, over 6 months (*Hyde, Bohlman & Valle, 2018*) hand-digitized all the transmission lines in the Legal Amazon from satellite data and compared them to two public datasets, which differed from each other. Only one was spatially accurate and neither dataset included every transmission line in the region. This demonstrates the importance of external validation of spatial and non-spatial accuracy to ensure data quality. One way to increase confidence in public datasets and reduce time spent validating data before use would be to develop a system allowing users to rate completeness and accuracy of these datasets.

## Caveats

While considerable effort was made to obtain a representative sample of stakeholders interested in all infrastructure types in the Amazon region, the survey population was biased toward a hydropower focus. However, we note that the literature review results were also skewed towards dams. Thus, this may simply reflect the general focus of the scientific community on dams in the Amazon region in the context of the hydropower boom, as the Amazon offers the greatest untapped hydropower potential in the country, yet dam development has extensive negative local social and ecological impacts (*Athayde et al., 2019*). Despite the focus on dams, at least one person searched for every attribute for every infrastructure type, so we believe that our results and the critical data attributes can be used broadly across infrastructure types. Finally, most of our survey respondents held academic positions, so these results may reflect the needs of the academic research community more strongly than those of government, the private sector, or NGO communities.

In this article, we assessed accessibility and quality for infrastructure data available at the federal level. However, government entities at smaller spatial scales, such as states and municipalities, are often responsible for the technical and administrative process of environmental impact assessment that generates relevant data. For example, in the Brazilian Amazon, states are responsible for the management, monitoring, inspection and even suspension of licenses for infrastructure projects. Quality of data therefore may vary among states and the lack of communication between the states, federal agencies and municipalities may result in incomplete data disclosure. Although beyond the scope of this article, investigation of state and municipal data provisioning and compliance is likely to explain some of the issues in data quality and accessibility found here and may reveal the need for greater investment in training, data infrastructure, and support at these government levels. Support (human and financial) for open data programs and institutional commitment to open data may also vary among agencies in the same country and lead to lower data accessibility and quality (*Tejedo-Romero, Ferraz Esteves Araujo & Gonçalves Ribeiro, 2025*).

## CONCLUSION AND FUTURE DIRECTIONS

Access to high quality data about infrastructure projects has the potential to improve the quality and efficiency of social-ecological research and assessment of impacts related to new and planned development projects. With access to the best data available, third parties and governments can help ensure the accuracy and accountability of environmental impact assessments (*Laurance et al., 2015*), fair compensation to impacted communities, and appropriate mitigation plans (*Hunter et al., 2021*). They can also inform improved watershed-level planning, as opposed to the project-by-project planning (*Athayde et al., 2019*), as well as better identify projects that are more harmful than helpful. Data transparency may also help reduce corruption in the infrastructure sector (*Kaufmann & Bellver, 2005*; *Ruijer, Grimmelikhuijsen & Meijer, 2017*).

To achieve these goals, governments and third parties would need to release datasets that conform to task-independent standards and contain, at minimum, the critical attributes identified in this study. In cases where governments have not or cannot provide

accessible and high-quality data, the research and NGO communities can be important sources of data to fill in these gaps. In addition, it is important that individual researchers share data on infrastructure that they collect on public archives. A culture of open data will reduce the redundant collection of data while allowing researchers to be credited for their work through citations (*Allen & Mehler, 2019*). The research and government communities should also strive to remove barriers to accessibility by investing in comprehensive and up-to-date central repositories to host this data. As countries continue to expand and update their infrastructure, promoting transparency and data sharing about the projects is an important step in implementing the right to access to information, as well as improved public participation in decision-making related to current and future infrastructure.

## ACKNOWLEDGEMENTS

We would like to thank the University of Florida's (UF) School of Forest, Fisheries and Geomatics Sciences and the UF Tropical Conservation and Development Program (TCD) as well as Dr. Stephen Perz, Dr. David Kaplan, and the 2015 Water Institute Graduate Fellows for their support. Dr. Charles Jekel was instrumental in helping with coding issues. We are grateful to our colleagues in the Amazon Dams International Research Network/ Rede Internacional de Pesquisa em Barragens Amazônicas/Red Internacional de Investigación en Represas Amazónicas (ADN/RBA/RIRA), to the participants of the civil society initiative and working group GT Infraestrutura, and to our associates at the Agência Nacional de Energia Elétrica (ANEEL) and Empresa de Pesquisa Energética (EPE) for their input and support during this research process. We are also grateful to the three anonymous reviewers whose suggestions improved our article. Any opinions, findings, and conclusions or recommendations expressed in this article are those of the authors and do not necessarily reflect NSF views.

### Funding

This research is based upon work supported by the U.S. National Science Foundation (NSF) under grant 1617413. Support was also provided by USDA National Institute of Food and Agriculture, McIntire Stennis project 1024612 to Stephanie A. Bohlman. Jacy L. Hyde and Christine Swanson were supported by the University of Florida Water Institute. Christine Swanson had additional funding from the University of Florida Informatics Institute, NASA FINESST award 80NSSC19K1355, and the NSF. Part of this research was performed while Christine Swanson held a National Research Council Research Award at the U.S. Naval Research Laboratory. There was no additional external funding received for this study. The funders had no role in study design, data collection and analysis, decision to publish, or preparation of the manuscript.

## Grant Disclosures

The following grant information was disclosed by the authors:

U.S. National Science Foundation (NSF): 1617413.

USDA National Institute of Food and Agriculture, McIntire Stennis: 1024612.

University of Florida Water Institute.

University of Florida Informatics Institute, NASA FINESST award 80NSSC19K1355, and the NSF.

National Research Council Research Award at the U.S. Naval Research Laboratory.

## Competing Interests

The authors declare that they have no competing interests.

## Author Contributions

- Jacy L. Hyde conceived and designed the experiments, performed the experiments, analyzed the data, prepared figures and/or tables, authored or reviewed drafts of the article, and approved the final draft.
- Christine Swanson analyzed the data, prepared figures and/or tables, authored or reviewed drafts of the article, and approved the final draft.
- Stephanie A. Bohlman conceived and designed the experiments, authored or reviewed drafts of the article, and approved the final draft.
- Simone Athayde conceived and designed the experiments, authored or reviewed drafts of the article, and approved the final draft.
- Emilio M. Bruna conceived and designed the experiments, authored or reviewed drafts of the article, and approved the final draft.
- Denis R. Valle conceived and designed the experiments, authored or reviewed drafts of the article, and approved the final draft.

## Human Ethics

The following information was supplied relating to ethical approvals (*i.e.*, approving body and any reference numbers):

The University of Florida's Institutional Review Board granted ethical approval to carry out the survey in this research study (B201600928).

## Data Availability

Data is available at Zenodo:

Hyde, J. L., Swanson, A. C., Bohlman, S. A., Athayde, S., Bruna, E. M., & Valle, D. R. (2024). A lack of open data standards for large infrastructure projects hampers social-ecological research in the Brazilian Amazon (1.1) [Data set]. Zenodo. https://doi.org/10.5281/zenodo.10626908.

Code is also available at Zenodo:

cswan5. (2024). cswan5/data_availability: Release for DOI (v1.0.1). Zenodo. https://doi.org/10.5281/zenodo.10627392.

## Supplemental Information

Supplemental information for this article can be found online at http://dx.doi.org/10.7717/peerj.19926#supplemental-information.

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
