# Peer review of "A lack of open data standards for large infrastructure projects hampers social-ecological research in the Brazilian Amazon"

_PeerJ, doi:10.7717/peerj.19926_

## Round 0.1 · original submission · Major Revisions

Thank you for submitting your manuscript. Three experts in the field have reviewed your paper, and based on their evaluations, we invite you to submit a revised version of your manuscript.

The reviewers have provided detailed comments and suggestions that you must address carefully. Please prepare:

1. A revised version of your manuscript incorporating the changes suggested by the reviewers.
2. A detailed point-by-point response document addressing each reviewer's comments. Please provide a clear scientific justification for any suggestions you choose not to implement.

Please note that submitting a revised version does not guarantee acceptance for publication. The revised manuscript will be re-evaluated by the same reviewers (if available) to ensure all concerns have been adequately addressed.

We look forward to receiving your revised manuscript.

Reviewer 1 ·

Basic reporting

The article is well-written, easy to understand and technically structured. The introduction is well-written and well-grounded in scientific literature.
The structure and logical sequence of the sessions follows a coherent narrative pattern, with the exception of session 3.3, which would be better placed after session 3.4.
The figures are somewhat disordered in the pdf and deserve careful revision with regard to numbering in the text.

Experimental design

The experimental design has an interesting basis in that it seeks three different types of hypothesis testing. There is also good results to be better explored.
However, at various times there seem to be some flaws and/or disconnects with the hypotheses tested and subsequently with the results presented (see more comments about the findings).
For example, about the experimental design- since data from other non-public sources is equally important for research, why did you evaluate only public datasets?
Another relevant issue is the accuracy of non-spatial data. Vetrò et al. (2016) suggest testing the accuracy of spatial components of the data and this is well publicized in the literature. How do you expect the accuracy of non-spatial components to be tested? This is not clear to the reader.
In general, the discussion on accuracy deserves a little more care (both in the methodology and in the results), since geographic accuracy depends on several factors, including temporal and thematic issues, and metadata is the key tool for determining it. Thus, it is not clear how independent a criterion such as standart machine readable is from the metadata, since projection and temporality information should be available in the metadata and somehow influence the ability to read the data.

Validity of the findings

The article presents many important and robust findings, but they contrast with assertions that do not necessarily derive from the results presented. Or they are partially derived and the conditioning factors or gaps should be better explained.

- The return rate of the survey was relatively low, with the majority of responses coming from a specific audience of researchers. Although this is only presented at the end in session 4.5, this information must also be reflected in some of the discussion earlier in the text, for example, with the conclusions being directed towards the use of this type of audience.

- One of the main findings is in session 3.3 regarding what is demanded and what is actually used, which could be explored further. What are these gaps and what positive implications would we have from having this other information available? This discussion only appears again in session 4.4, based on the findings of other articles, and does not present anything new related to the research carried out.

- Another important finding (1st paragraph, section 3.2.2/ and line 288): the data is available and was mostly found in the search process! This seems to reflect a different discussion (line 363). The sentence is partly in line with what the results indicated, since:
- 95% of respondents were able to find data
- Of the 188 searches for government data, 82% of the datasets were found in government sources
- In the assessment, the datasets had high spatial accuracy (96% and 98% for dams and roads), indicating the spatial reliability of the data

-Lines 302-304: the data could be better interpreted if they were presented as a proportion of the total analyzed

- Lines 353- 355 “extensive, time-consuming searches” “low quality”: does not seems to be the direct conclusion that your research drew. The results clearly indicate a bimodal approach to search time and data processing, which make this sentence seems to be a hasty conclusion. Perhaps what can be brought up here is how the absence of metadata may compromise data quality (which could be a different way of presenting your findings). At the end, it is not clear for what the interviewed users spend additional time on formatting data. In this respect, this finding deserves a better discussion, since working on data formatting is much more inherent to the process of doing research than necessarily the quality of the data made available. The way in which the research was carried out does not allow us to draw this kind of conclusion. Modeling processes, often used for impact assessment, require a great deal of time to process spatial data.

- About the critical atributes proposal (section 3.3): how different or similar (close or far) it is from your findings (both survey, literature review and your data evaluation)? How could future research be improved? What additional results could we expect from research that finds these attributes available? This section would be better positioned after section 3.4.

- Section 3.3: Although spatial extent is a frequently requested data, point location is very often used. Why you decide to do not have it here as the recommended minimum atributes, and also as a result of your research?

- Lines 363-364: This is not entirely in line with your findings, since a bimodal distribution of search time was found. A suggestion would be to bring to this session a discussion on how this search time and bimodal distribution could be improved if there were unified database repositories

- Line 373 - “12 abandoned projects”: it would be better to give the proportion here and improve the discussion. Many projects are in fact abandoned (e.g. resources, deadlines, human resources), and it is not clear whether this number/proportion is part of an usual statistic or whether it is in fact due to a lack of data.

- Super importante conclusions and well explored could be seen at lines 399-402

·

Basic reporting

Consider uniformizing the way you refer to attributes throughout the text and in figures/tables (e.g. Complete/completeness; currency/currentness).

Experimental design

No comment

Validity of the findings

Even if the purpose of the paper is to investigate availability for research questions, consider whether Project owner; funding and budget should not be included among the priority attributes, since they are key in terms of accountability and corporate social responsibility. Project owner information may allow for social control over infrastructure investments. Information on funding sources is also particularly important for co-responsibility of investment angencies and companies regarding impacts on biodiversity and communities. Environmental and social NGOs could work with multilateral investors in order to push governments to improve infrastructure projects data quality and availability.

·

Basic reporting

The article conforms to professional standards of courtesy and expression.
The article did not include information in the introduction to contextualize the Brazilian Amazon.
The figures, tables, and raw data were correctly provided.
Authors should explore the results to better understand how data is provided by states in the Brazilian Amazon.

Experimental design

Comments are in the attached file.

Validity of the findings

Comments are in the attached file.

---

## Round 0.2 · accepted · Accept

Thank you for your revised manuscript. I've assessed it with one of the previous reviewers and am happy with the changes you made. Your manuscript is ready for publication! Please work with our production team for the next steps.

·

Basic reporting

No comment

Experimental design

No comment

Validity of the findings

No comment